# Education Level and Self-Reported Cardiovascular Disease in Norway—The Tromsø Study, 1994–2016

**DOI:** 10.3390/ijerph20115958

**Published:** 2023-05-25

**Authors:** Celina Janene Cathro, Tormod Brenn, Sairah Lai Fa Chen

**Affiliations:** Department of Community Medicine, UiT—The Arctic University of Norway, 9019 Tromsø, Norway

**Keywords:** cardiovascular disease, education, heart attack, stroke, angina, socioeconomic status, logistic regression, Tromsø Study, Norway

## Abstract

Background: Cardiovascular disease (CVD) is a leading source of morbidity and mortality, and research has shown education level to be a risk factor for the disease. The aim of this study was to investigate the association between education level and self-reported CVD in Tromsø, Norway. Methods: This prospective cohort study included 12,400 participants enrolled in the fourth and seventh surveys of the Tromsø Study (Tromsø4 and Tromsø7) in 1994–1995 and 2015–2016, respectively. Logistic regression was used to obtain odds ratios (ORs) and 95% confidence intervals (CIs). Results: For every 1-level increase in education, the age-adjusted risk of self-reported CVD decreased by 9% (OR = 0.91, 95% CI: 0.87–0.96), but after adjustment for covariates, the association was weaker (OR = 0.96, 95% CI: 0.92–1.01). The association was stronger for women (OR = 0.86, 95% CI: 0.79–0.94) than men (OR = 0.91, 95% CI: 0.86–0.97) in age-adjusted models. After adjustment for covariates, the associations for women and men were similarly weak (women: OR = 0.95, 95% CI: 0.87–1.04; men: OR = 0.97, 95% CI: 0.91–1.03). In age-adjusted-models, higher education level was associated with a lower risk of self-reported heart attack (OR = 0.90, 95% CI: 0.84–0.96), but not stroke (OR = 0.97, 95% CI: 0.90–1.05) or angina (OR = 0.98, 95% CI: 0.90–1.07). There were no clear associations observed in the multivariable models for CVD components (heart attack: OR = 0.97, 95% CI: 0.91–1.05; stroke: OR = 1.01, 95% CI: 0.93–1.09; angina: OR = 1.04, 95% CI: 0.95–1.14). Conclusions: Norwegian adults with a higher education level were at lower risk of self-reported CVD. The association was present in both genders, with a lower risk observed in women than men. After accounting for lifestyle factors, there was no clear association between education level and self-reported CVD, likely due to covariates acting as mediators.

## 1. Introduction

Cardiovascular disease (CVD) is a leading source of morbidity and mortality in Norway as well as globally [1,2]. There are several factors that can impact the risk for CVD, including diabetes, unhealthy diet, overweight, physical inactivity, smoking, high cholesterol, blood pressure, alcohol consumption [3], and stress [4]. A Norwegian study found that coronary heart disease incidence declined between 1995 and 2010 amongst Tromsø Study participants. This decline was attributed to decreases in annual hospitalisations for ST-segment-elevation myocardial infarction, out-of-hospital sudden deaths, and decreases in CVD risk factors, such as smoking, high blood pressure, cholesterol, and physical inactivity [5]. Yet, CVD prevalence is expected to increase globally due to the aging population and population growth [2]. The most common CVDs in Norway are angina, heart attack, heart failure, and stroke [6].

Social standing or class, termed socioeconomic status (SES), is measured by one or a combination of education level, occupation, and income [7]. The more education people have, the more likely they are to make informed decisions regarding their health, as those with less education are more likely to have a lower level of health literacy [7,8] Those with a high education level are also more likely to gain secure employment and have higher earnings, allowing them the means to afford a healthier lifestyle and adequate healthcare [7]. Education level is a known predictor of CVD morbidity and mortality risk [8,9,10,11,12,13,14,15]. Interestingly, a South Korean study found that of the different SES variables analysed, educational level was the only variable associated with an increased risk of major cardiac events for patients who underwent a percutaneous cardiac intervention after an acute myocardial infarction [16]. Neighbourhood of residence can also play a role in CVD risk, as poorer neighbourhoods often have fewer grocery stores than wealthier areas, resulting in the consumption of more ready to go foods [17]. In fact, a recent Canadian study found that life expectancy can vary by more than 10 years for men living in census tracts 5 kilometres apart from each other within the center of Vancouver [18]. Families with a low SES are also more likely to become obese, develop hypertension, and multimorbidities [17,19].

A 2014 Norwegian study reported that incident acute myocardial infarctions are more common among people with a compulsory education level [12]. The study had split education into three levels: basic (compulsory education), upper secondary (high school/vocational school), and tertiary (college/university) [12]. According to a review on SES and CVD, not only are people with a low education level at higher risk for a cardiac event, but they are also more likely to have less optimal short- and long-term outcomes after a cardiac event [8]. In addition, a study analysing national register data for populations aged 65 years and older in Norway, Sweden, Finland, and Denmark from 2001 to 2015 found that those with a high education level consistently had higher life expectancies than those with a low education level [20]. Currently, education and healthcare are very accessible to the Norwegian population because higher education is tuition-free [21], and the country has a universal healthcare system [22]. To further elucidate the relationship between measures of SES and health in Norway, this study aimed to investigate the association between education level and self-reported CVD in Tromsø, Norway.

## 2. Materials and Methods

### 2.1. Study Design, Questionnaires, and Clinical Visits

The Tromsø Study is an ongoing population-based study conducted in Tromsø, Norway; it began in 1974 and includes over 45,000 participants recruited across seven surveys [23,24]. The present study includes information from the fourth survey of the Tromsø Study (Tromsø4) in 1994–1995 and the seventh survey (Tromsø7) in 2015–2016. All men and women residing in Tromsø aged 25 or older were invited to participate in Tromsø4, with a response rate of 69.6% (n = 12,865) and 74.9% (n = 14,293), respectively. In Tromsø7, all men and women aged 40 or older were invited to participate, with a response rate of 62.4% (n = 10,009) and 67.0% (n = 11,074), respectively. Participants attended a clinical visit at Tromsø4 and completed a questionnaire at Tromsø4 and Tromsø7. The questionnaires collected information on education level, CVD components (heart attack, stroke, and angina), lifestyle factors, and other variables [25,26]. During the clinical visit, height, weight, and systolic blood pressure (SBP) were recorded, and a blood sample was collected.

### 2.2. Study Sample

Of the 12,867 participants who took part in both Tromsø4 and Tromsø7, we excluded those with the following characteristics at Tromsø4: over 70 years of age, missing information on education level, reported any of the CVD components, or had missing information on any of the CVD components. After these exclusions, 12,400 participants remained. Participants with missing information on a CVD component at Tromsø7 were excluded from the analyses on that component (Figure 1). For any CVD, participants with missing information on any of the CVD components were excluded from the analyses unless they had answered yes to at least one CVD component.

### 2.3. Assessment of Education Level, Covariates, and Cardiovascular Disease

Data on education level was assessed from the Tromsø4 questionnaire [26]. Participants selected the highest education level they had completed (Table 1).

Information on gender (women, men), age group (25–29, 30–39, 40–49, 50–59, 60–69), light physical activity (i.e., not sweating or out of breath: 0, <1, 1–2, ≥3 h/week), hard physical activity (i.e., sweating/out of breath: 0, <1, 1–2, ≥3 h/week), alcohol consumption (times/month, 0 if less than once per month, low-alcohol beer not included), daily cigarette smoking (yes, no), and previous or current diabetes (yes, no) was also assessed from the Tromsø4 questionnaire [26]. Body mass index (BMI, <25, ≥25 and <30, or ≥30) and SBP (mmHg) were measured during the Tromsø4 clinical visit. Three SBP readings were taken, and the reported SBP was calculated as the mean of the second and third readings. Finally, the blood samples from Tromsø4 were used to determine total cholesterol (mmol/L).

Information on the CVD outcome, was assessed from the Tromsø7 questionnaire using the following questions: “Have you ever had, or do you have?”: heart attack (no; previously, not now); cerebral stroke/brain haemorrhage (no; previously, not now); angina pectoris (heart cramp) (no; yes, currently; previously, not now) [25]. If participants had answered yes to any of these diseases (heart attack, angina, and/or stroke), any CVD was also taken as a yes.

### 2.4. Statistical Analysis

To estimate the association between education level and self-reported CVD, binary logistic regression was used to obtain odds ratios (ORs) with 95% confidence intervals (CIs). Age-adjusted and multivariable logistic regression were performed for any self-reported CVD combined, and for the CVD components heart attack, stroke, and angina separately, with education level included as an ordinal variable (with values 1–5) and in separate analyses in categories to assess the effect of each educational level. The analyses were also performed separately for gender.

Inclusion of covariates in the multivariable models were decided by including potential candidate variables initially and removing those variables that were statistically insignificant. Thereby, all multivariable models included gender, age, daily cigarette smoking, SBP, and cholesterol. For self-reported heart attack, alcohol consumption and BMI were additionally included. IBM SPSS Statistics Version 28 was used to perform all statistical analyses [27].

## 3. Results

A total of 12,400 participants were included in the study (n = 6673 women, n = 5727 men). The most commonly reported education level was EL2 (30.0%), followed by EL1 (25.0%), EL4 (17.8%), EL5 (16.9%), and EL3 (10.3%). The least common education level among those 25–29 years old was EL1 (13.3%), and the most common was EL2 (28.0%).The least common education level among those 60–69 years old was EL3 (4.2%), and the most common was EL1 (49.1%). The mean monthly alcohol consumption increased with increasing education level (2.3 times/month for EL1; 4.8 times/month for EL5). The proportion of participants that were daily smokers was highest in EL2 (34.1%) and lowest in EL5 (8.8%). The proportion of participants with BMI ≥ 30 was highest in EL1 (34.4%) and lowest in EL3 (10.0%). Mean SBP and total cholesterol were lowest for those in EL3 (SBP = 126.3 mmHg, total cholesterol = 5.5 mmol/L) and highest for those in EL1 (SBP = 131.9 mmHg, total cholesterol = 6.2 mmol/L) (Table 2).

The number of self-reported cases and results from the age-adjusted and multivariable models with education included as an ordinal variable and in categories are displayed in Table 3 and Table 4, respectively. The number of self-reported cases were 1024, 542, 366, and 261 for CVD, heart attack, stroke, and angina, respectively. There was a negative association between education level and self-reported CVD in the age-adjusted model with education level included as an ordinal variable (OR = 0.91, 95% CI: 0.87–0.96). In the multivariable model for the education level ordinal variable and any self-reported CVD the OR was 0.96 (95% CI: 0.92–1.01). The OR for the education level ordinal variable and self-reported heart attack was 0.90 (95% CI: 0.84–0.96) in the age-adjusted model and 0.97 (95% CI: 0.91–1.05) in the multivariable model. For self-reported stroke, the values were 0.97 (95% CI: 0.90–1.05) and 1.01 (95% CI: 0.93–1.09), respectively. For self-reported angina, the values were 0.98 (95% CI: 0.90–1.07) and 1.04 (95% CI: 0.95–1.14), respectively.

The number of CVD cases was approximately two times higher in men compared to women (Table 3). The association was stronger for women (OR = 0.86, 95% CI: 0.79–0.94) than men (OR = 0.91, 95% CI: 0.86–0.97) in age-adjusted models with education level included as an ordinal variable. However, after adjustment for covariates, the ORs for women became more similar to the ORs for men, and no statistically significant association remained for either gender. The categorical inclusion of educational level did not give a strong indication of non-linearity, although the relationship did not appear to be a perfect dose-response one (Table 4).

## 4. Discussion

Overall, there was a wide distribution of education levels in this population of adult Norwegians, with the most common being EL2 (i.e., technical school, middle school, vocational school, 1–2 years senior high school). We observed that a high education level was associated with lower any self-reported CVD risk, as well as a lower risk of self-reported heart attack in age-adjusted models.

Our observations are consistent with a large body of existing literature that has reported a negative association between education level and CVD outcomes [8,10,12,13,28]. A possible explanation for this association is the ability to make better informed health decisions with increasing education level [29]. An American systematic review reported that an average of 39% of the heart failure patients included in their study had poor health literacy skills [29]. It also reported a positive association between health literacy and medication adherence. A German cross-sectional study found that having a problematic or insufficient level of health literacy was independently associated with CVD and healthcare use in comparison to participants with a sufficient level of health literacy [30]. Additionally, a meta-analysis found that not only was there a positive and statistically significant correlation between health literacy and patient adherence to medical treatment for both chronic and acute illnesses, but also that health literacy interventions were an effective means of increasing both health literacy and patient adherence [31].

When we applied age-adjusted models in men and women separately, negative associations remained between education level and self-reported CVD and heart attack. However, the protective associations were stronger for women. These results are consistent with previous findings, which also showed discrepancies between the genders [32]. A meta-analysis and systematic review, which included over 22 million participants from 116 cohorts, reported a stronger negative association between education level and CVD, and between education level and coronary artery disease, in women than in men [32]. The results also showed a 24% and 18% greater excess risk of coronary heart disease and CVD, respectively, for women than men (when comparing lowest level of education to the highest) [32]. Additionally, a study using data from surveys 2 through 6 of the Tromsø Study observed that men had a higher risk of myocardial infarction throughout life after adjustment for risk factors [33]. One potential reason for this difference is that women are disproportionately affected by poverty and therefore more susceptible to the ill health and poor quality of life that can result from lower income [8].

In the multivariable models, there were no clear associations between education level and any self-reported CVD, heart attack, stroke, and angina. Contrary to our findings in the multivariate model, a systematic review and meta-analysis of 72 cohorts across Europe, America, and Asia (including only studies that adjusted for covariates in the pooled analysis), concluded that overall, lower levels of education were associated with an increased risk for cardiovascular outcomes [13]. On the other hand, Woodward et al. [15] compared the relationships between education and CVD in Asian and Australasian populations using 24 cohort studies. This comparison adjusted for age, sex, SBP, total cholesterol, BMI, smoking, and alcohol consumption [15]. In the high-income Australasian populations, the findings were similar to our study. That is, in the adjusted model for all CVD (fatal or non-fatal), the findings were statistically insignificant for both secondary (1.04 (0.92 to 1.18)) and primary or none (1.11 (0.99 to 1.25)) in comparison to tertiary education [15].

A possible explanation as to why the protective association in self-reported CVD risk disappears after adjusting for covariates could be that many covariates serve as mediators [34]. In this study, the adjustment for covariates could have accounted for the effect of the mediator variables in the relationship between education level and CVD. Part of the reason why many of these variables could be acting as mediators is because higher education can enable more job opportunities and the potential for higher income [15], which is one of the indicators of SES and is also associated with lower CVD risk [8,10,17]. As healthy diets tend to cost more in Western countries, lower income families may not be able to afford nutritious diets rich in fruits and vegetables [17]. Research also shows that in high income countries, people of a lower SES are more likely to smoke [10,15]. Furthermore, similar to the findings by Woodward et al. [15] amongst Australasians, we found that on average, the more educated drank more than their less educated counterparts. On the other hand, the review by Psaltopoulou et al. [17] found that those of a lower SES were more likely to drink in excess. Research has shown that the relationship between alcohol consumption and heart health follows a J-shaped curve, meaning moderate alcohol consumption is associated with a decreased risk of cardiovascular events, while excessive drinking increases this risk [17,35]. A Danish descriptive cross-sectional study found that people of a higher SES were more likely to drink wine, which was associated with optimal functioning in terms of personality, psychiatric symptoms, and health-related behaviours in comparison to those of a lower SES [36]. Beer on the other hand, was associated with suboptimal functioning for the same variables [36]. Interestingly, a recent British study concluded that education level was inversely associated with both mental health problems and CVD [37]. The researchers also found that depression mediated 2% of the inverse association found between education level and CVD [37]. This explanation reflects the conclusions of several other studies: that various risk factors mediate the effects of education level on CVD [38,39,40]. For example, a Dutch study found that the inverse association between coronary heart disease and education level was mediated by biological and behavioural variables such as smoking, obesity, physical inactivity, and hypertension [39]. Furthermore, a recent study using data from Tromsø7 found that the statistically significant association between area-level SES and CVD risk factors was mediated by lifestyle factors (smoking, alcohol, and physical activity) [41]. Physical activity was one among many of the potential variables we included initially as a potential covariate. However, the variable was not included in our multivariable analyses, as the effect was statistically insignificant. This was contrary to the protective association observed between higher physical activity levels and lower CVD risk in previous studies [42,43] and given the above findings from Tromsø7. However, Tromsø7 investigated overall SES, and the outcomes were not identical to ours [41]. In our study, there appears to be a mediating effect of covariates on the association between education level and CVD. Findings from this study thus may be of use to future researchers investigating differences in the relationship between education level and CVD outcomes on a global scale. We would recommend for future researchers to investigate the effectiveness of strategies that could be implemented to help attenuate the differences for CVD risk between individuals with different education levels and even between genders. Further investigation of this relationship could provide insight for the design of future targeted preventative programs. This should be assessed using formal mediation analysis techniques.

The present study has several strengths, including its population-based design with high participation, standardised methodology, and repeated data collection. This study also has some weaknesses. Since a large proportion of the data was self-reported, response and recall bias may be present [44]. However, a study on the validity of self-reported stroke in Tromsø4 concluded that it is acceptable to use questionnaires to assess previous stroke [45]. Additionally, a study on the validity of self-reported education level in Tromsø7 concluded that this data is adequate for research use [46]. On the other hand, the study also found that the associations between education level and cardiometabolic disease may be weaker when using data from Tromsø7 than when using registry-based data [46]. Lastly, due to the study design, differential loss-to-follow-up may have occurred, thus biasing the results toward the null [44].

## 5. Conclusions

Among adult Norwegian men and women, higher education level was associated with a lower risk of self-reported CVD. The association was present in both genders, with greater risk reductions observed in women. However, when lifestyle factors and other participant characteristics were considered in multivariable models, there were no clear associations between education level and any self-reported CVD. The weakened associations suggest that daily cigarette smoking, alcohol consumption, BMI, SBP, and total cholesterol may have played a mediating role. As CVD prevalence is expected to increase globally [2], it is important for governments to support and review research on socioeconomic inequalities in health and disease, so that plans to help mitigate future illness can be implemented as early as possible.

## Figures and Tables

**Figure 1 ijerph-20-05958-f001:**
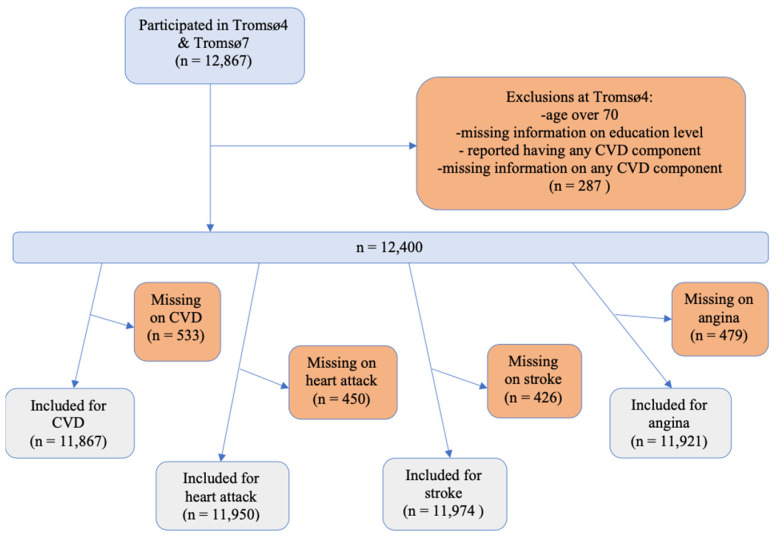
Flowchart of sample selection. Any exclusions below the blue line (n = 12,400) occurred at Tromsø7.

**Table 1 ijerph-20-05958-t001:** Abbreviations and answer choices for the highest obtained education level in Tromsø4 [26].

Abbreviation	Highest Obtained Education Level
EL1	7–10 years primary/secondary school, modern secondary school
EL2	Technical school, middle school, vocational school, 1–2 years senior high school
EL3	High school diploma (3–4 years)
EL4	College/university, less than 4 years
EL5	College/university, 4 or more years

**Table 2 ijerph-20-05958-t002:** Mean (standard deviation) and percentages within each education level, Tromsø4, 1994–1995 ^1^.

Education Level ^2^					
	EL1	EL2	EL3	EL4	EL5
N (%)	3106 (25.0)	3717 (30.0)	1271 (10.3)	2213 (17.8)	2093 (16.9)
Gender					
Women	26.8	28.4	11.4	16.7	16.7
Men	23.0	31.8	8.9	19.2	17.1
Age (years, %)					
25–29	13.3	28.0	21.9	20.7	16.1
30–39	17.5	31.5	13.5	19.5	18.0
40–49	26.9	30.1	6.5	17.4	19.2
50–59	39.2	29.0	3.9	14.5	13.4
60–69	49.1	27.1	4.2	13.1	6.5
Alcohol consumption (times/month)	2.3 (2.8)	2.8 (3.0)	2.9 (3.2)	3.6 (3.6)	4.8 (4.6)
Daily cigarette smoking (%)	32.7	34.1	10.3	14.1	8.8
BMI (kg/m^2^, %)					
<25	22.0	29.2	11.0	18.6	19.2
≥25<30	27.8	31.3	9.1	17.2	14.6
≥30	34.4	29.8	10.0	15.6	10.1
SBP (mmHg)	131.9 (16.3)	129.5 (15.5)	126.3 (14.2)	128.0 (15.4)	126.8 (14.2)
Total cholesterol (mmol/L)	6.2 (1.2)	5.9 (1.2)	5.5 (1.1)	5.6 (1.2)	5.6 (1.1)

^1^ Some numbers are smaller due to missing values, especially for alcohol consumption (7.7%). ^2^ See Table 1 for education level categories. Abbreviations: BMI: body mass index, SBP: systolic blood pressure.

**Table 3 ijerph-20-05958-t003:** Associations between education level (see Table 1 for levels) included as an ordinal variable (1–5) and self-reported CVD, heart attack, stroke, and angina. The Tromsø Study, 1994–2016.

	Age-Adjusted		Multivariable	
	Cases/n	OR (95% CI)	Cases/n	OR (95% CI)
Any CVD	1024/11,867	0.91 (0.87–0.96)	1023/11,825 ^1^	0.96 (0.92–1.01)
Women	343/6341	0.86 (0.79–0.94)	343/6320	0.95 (0.87–1.04)
Men	681/5526	0.91 (0.86–0.97)	680/5505	0.97 (0.91–1.03)
Heart attack	542/11,950	0.90 (0.84–0.96)	488/11,000 ^2^	0.97 (0.91–1.05)
Women	144/6395	0.77 (0.67–0.88)	125/5777	0.89 (0.76–1.04)
Men	398/5555	0.91 (0.85–0.98)	363/5223	1.00 (0.92–1.09)
Stroke	366/11,974	0.97 (0.90–1.05)	365/11,932 ^1^	1.01 (0.93–1.09)
Women	143/6405	0.94 (0.83–1.06)	143/6384	1.01 (0.89–1.15)
Men	223/5569	0.97 (0.88–1.06)	222/5548	1.00 (0.91–1.10)
Angina	261/11,921	0.98 (0.90–1.07)	261/11,879 ^1^	1.04 (0.95–1.14)
Women	105/6381	0.92 (0.79–1.06)	105/6360	1.02 (0.88–1.19)
Men	156/5540	1.02 (0.91–1.14)	156/5519	1.08 (0.96–1.21)

^1^ Adjusted for gender, age, daily cigarette smoking, SBP, and total cholesterol. ^2^ Adjusted for gender, age, alcohol consumption, daily cigarette smoking, BMI, SBP, and total cholesterol. Abbreviations: BMI: body mass index, CI: confidence interval, CVD: cardiovascular disease, OR: odds ratio, SBP: systolic blood pressure.

**Table 4 ijerph-20-05958-t004:** Associations between education level (see Table 1 for levels) in categories and self-reported CVD, heart attack, stroke, and angina. The Tromsø Study, 1994–2016.

	Age-Adjusted		Multivariable	
	Cases/n	OR (95% CI)	Cases/n	OR (95% CI)
Any CVD	1024/11,867		1023/11,825 ^1^	
EL1	332/2894	1.00 (Ref)	332/2883	1.00 (Ref)
EL2	343/3551	1.07 (0.91–1.26)	342/3538	1.08 (0.91–1.28)
EL3	60/1235	0.73 (0.54–0.98)	60/1229	0.83 (0.61–1.12)
EL4	171/2150	0.93 (0.76–1.14)	171/2144	1.01 (0.82–1.24)
EL5	118/2037	0.66 (0.53–0.82)	118/2031	0.83 (0.66–1.05)
Heart attack	542/11,950		488/11,000 ^2^	
EL1	185/2929	1.00 (Ref)	158/2569	1.00 (Ref)
EL2	181/3577	1.01 (0.82–1.26)	169/3309	1.05 (0.83–1.33)
EL3	28/1246	0.62 (0.41–0.93)	24/1161	0.69 (0.44–1.08)
EL4	92/2159	0.91 (0.70–1.18)	82/1924	0.98 (0.73–1.32)
EL5	56/2039	0.58 (0.43–0.79)	55/1924	0.91 (0.65–1.28)
Stroke	366/11,974		365/11,932 ^1^	
EL1	107/2935	1.00 (Ref)	107/2924	1.00 (Ref)
EL2	124/3580	1.23 (0.94–1.60)	123/3567	1.22 (0.93–1.60)
EL3	25/1245	1.00 (0.64–1.58)	25/1239	1.09 (0.69–1.72)
EL4	62/2163	1.08 (0.78–1.50)	62/2157	1.13 (0.81–1.57)
EL5	48/2051	0.89 (0.62–1.26)	48/2045	1.04 (0.73–1.49)
Angina	261/11,921		261/11,879 ^1^	
EL1	84/2924	1.00 (Ref)	84/2913	1.00 (Ref)
EL2	77/3555	0.96 (0.70–1.32)	77/3542	1.00 (0.73–1.39)
EL3	17/1239	0.85 (0.50–1.46)	17/1233	1.00 (0.58–1.73)
EL4	46/2161	1.02 (0.70–1.47)	46/2155	1.16 (0.79–1.69)
EL5	37/2042	0.87 (0.58–1.29)	37/2036	1.16 (0.77–1.74)

^1^ Adjusted for gender, age, daily cigarette smoking, SBP, and total cholesterol. ^2^ Adjusted for gender, age, alcohol consumption, daily cigarette smoking, BMI, SBP, and total cholesterol. Abbreviations: BMI: body mass index, CI: confidence interval, CVD: cardiovascular disease, OR: odds ratio, SBP: systolic blood pressure.

## Data Availability

Not applicable.

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
