# Peer review of "Education Level and Self-Reported Cardiovascular Disease in Norway—The Tromsø Study, 1994–2016"

_ijerph, 2023, doi:10.3390/ijerph20115958_

Round 1

Reviewer 1 Report

The paper is very interesting and well written. However, I would like to suggest the following.

1. In all multivariate analyses, there did not appear to be a significant association between education level and CVD's. It is evident that the relationship between education and CVD's that appears in the age-adjusted analyzes is mediated by the effect of lifestyle variables. The authors should explain why,therefore, the association between education level and CVD's is of interest.

2. Physical exercise should also be included as a covariate in the multivariate analyses. Regardless of whether or not the variable is significant in the bacward elimination model, physical activity must be controlled for its potential confounding  on the relationship of education and CVD's . Moreover, I think that the selection of lifestyle variables by the backward method is not needed. All variables that are potential confounders should be controlled for in the final multivariate model. 

3. The term multivariable ordinal model (or age-adjusted ordinal model) mentioned in various parts of the article (e.g. line 145, line 147) should be replaced with the correct one. It is more appropriate to clarify that the education variable is used as an ordinal rather than that the model is ordinal. Otherwise, it creates confusion.

Reviewer 2 Report

Manuscript review response: “Education level and self-reported cardiovascular disease in Norway – The Tromsø Study, 1994-2016”.

 Manuscript ID: ijerph-2380089

 Comments

It is an interesting article that finds a relationship between two important aspects of human development healthy growth and education (cardiovascular diseases and level education), I consider that it provides the necessary methodology to consider the relationship that exists between the environment where we do develop (epigenetic aspects) and the disease onset.

Line 36 you need mentioned factors such as lifestyle, diet, and physical inactivity among others.

Line 44 could you add: what the meaning of a low education level and high education level?

Line 45-47 I think it would be important to mention the relationship between socioeconomic level and stress, as well as the influence of the latter on the appearance of any cardiovascular problem.

Line 52-54 could you add what are the cardiovascular diseases more common in Norway?

Introduction

i suggest that you included the possible biological mechanism for which educational level impact on health (neurotransmission secretion low or high, behaviour-associated proteins and genes), and if this could be considered an epigenetic factor.

Discussion

Line 172 you could information in the discussion about association between education level and CVD mortality index

Line 193-194 You could add information on the hormone’s role in women for the development of CVD.

Line 221-222 could add the future perspectives of this protocol obtained from the results of your study.

 Conclusions

It would be important to point out why the level of study and its relationship with the socioeconomic level that affects the appearance of cardiovascular diseases, as well as the risk generated by the lack of information on cardiovascular diseases and the triggers of these.

English language is good
